# The Role of CSR Information on Social Media to Promote the Communicative Behavior of Customers: An Emotional Framework Enriching Behavioral Sciences Literature

**DOI:** 10.3390/bs13020126

**Published:** 2023-02-02

**Authors:** Zhijuan Li, Muhammad Safdar Sial, Hualiang Wu, Sorinel Căpușneanu, Cristian-Marian Barbu

**Affiliations:** 1Department of Economics and Management, Wuhan University, Wuhan 430072, China; 2The Center for Economic Development Research (CEDR), Wuhan University, Wuhan 430072, China; 3Department of Management Sciences, COMSATS University Islamabad, Islamabad 44000, Pakistan; 4Audit Department, Wuhan Huazhong University of Science and Technology Asset Management Co., Ltd., Wuhan 430072, China; 5Department of Economic Sciences, Titu Maiorescu University, 004042 Bucharest, Romania; 6Faculty of Management-Marketing, Artifex University of Bucharest, 060754 Bucharest, Romania

**Keywords:** CSR, advocacy behavior, customer–company identification, banking sector, gratitude

## Abstract

Studies have shown that an organization’s corporate social responsibility (CSR) activities affect customer behaviors such as loyalty and satisfaction. In spite of this, the role of social media in informing customers about a brand’s CSR activities and in fostering customer advocacy behavior (CADB) has been underexplored. To fill this knowledge gap, this study investigates the relationship between the CSR-related communication of a banking organization and CADB. This study also examines how emotions such as customer–company identification (CCI) and gratitude as a mediator and a moderator. Using a self-administered questionnaire (*n* = 302), we collected data from banking customers. Hypotheses were evaluated by using structural equation modeling, which revealed that CSR positively predicts CADB, whereas there are mediating and moderating functions of CCI and GA. Theoretically, this study highlights the role of human emotions in behavior formation from the standpoint of social media. Practically, this study provides important insights for the banking sector’s administrators to realize the important role of CSR communication, using different social networking websites, for converting customers into brand advocates.

## 1. Introduction

The escalating persistent use of social media by individuals worldwide has significantly transformed the way contemporary organizations set their marketing strategies in an online context [1]. Additionally, various organizations have started using different social networking websites, for example, Facebook, Twitter, etc., to effectively communicate with stakeholders, including customers [2]. Nowadays, social networks have gained popularity in real-world applications. So, its structure becomes more and more complex and enormously large. A social network can be seen as a graph of social individuals and the relationship between them, where the social individuals represent the vertices, and the connections between individuals constitute the edges of the graph [3]. Additionally, social networks influence diffusion and play a key role in finding efficient and effective ways to propagate information [4].

Unlike traditional media, the interactive communicability of social media undoubtedly makes it an effective forum of communication for companies in this digital age. The importance of social media can be estimated by observing the rising statistics of social media users globally. Furthermore, statistics indicate that over 4.5 billion people use social networking websites [5], and the number of social media users worldwide is increasing at a rate of 12% annually. Facebook alone has almost 3 billion active users, which shows the popularity of this social networking forum among individuals on a global landscape [5]. Perhaps these are the reasons why companies in the current age actively use social media for an effective marketing communication strategy and for building a relationship with customers [6,7]. At the same time, many organizations have developed business pages on different social networking websites to effectively communicate with customers and other stakeholders. According to a recent report, almost 200 million businesses use Facebook tools to communicate interactively with customers [8]. The world’s leading brands, such as Nike, TOMS, GoPro, Starbucks, Spotify, and Pop-Tarts, all ensure their active presence on different social media forums to communicate with their interest groups, including customers, effectively. Without a doubt, in the current digital age, social media not only facilitates companies to promote their products and services but also enables companies to engage effectively with customers, creating an opportunity for contemporary businesses to develop meaningful customer–brand relationships.

The reason companies desire to build long-term relationships with their customers lies in the increasing role of customers in bringing in organizational excellence for a company because, without the support of customers, an organization cannot be successful, especially in the long run [9,10]. Although the critical role of customers in the success of an organization has been highlighted in the literature previously, attracting and retaining customers to be repeat buyers of a brand is not easy. Equipped with digital technologies and better knowledge, modern customers have an increased level of expectations from companies because they not only observe the product and service delivery standards of an organization but also evaluate the corporate social responsibility (CSR) initiatives of a brand to benefit the ecosystem and society [11,12,13,14]. Buttering this, the seminal work by Sun et al. [15] indicates how companies use CSR as a strategic tool to promote green consumption. Considering the critical role of customers in the success of a brand, and considering that customers in the current age are interested to see the CSR engagement of a brand, companies use social media not only for promotional purposes but also to make the customer aware of different pro-social activities [16,17,18,19]. Undoubtedly, social media’s interactive and customized interface enables companies to engage with their customers and other stakeholders to solidify different CSR policies.

It is undoubtedly true that social media plays a crucial role in engaging customers with a brand, especially from the perspective of CSR. However, there exists a critical gap in this literature stream. That is, to date, a plethora of studies related to CSR activities of companies and communicating these activities with customers on social media attempted to spark customers’ behavioral intentions regarding loyalty [20], satisfaction [21], or purchase preferences [22], leaving a very important communicative behavior unattended. The literature names this communicative behavior as customer advocacy behavior (CADB), which is defined as the behavior of customers who not only support a company’s product or services but also promote it among their social circles [23]. Additionally, customers tend to defend their favorite brand when they see any criticism from the detractors. We tend to initiate this debate on the CSR–CADB relationship in a social media context due to two specific reasons. First, empirical evidence suggests that customers prefer user-generated communication over company-generated marketing communication prior to finalizing their purchase decision regarding a product or a brand [1,24]. This clearly highlights the critical role of customers’ communicative behavior in the forms of referrals, endorsements, and positive word of mouth to grab the attention of other customers to prefer a specific brand during their purchase decisions. This argument has also been emphasized by Sweeney et al. [25]. Even some researchers rate the communicative behavior of customers higher than loyalty and purchase intentions [26,27]. Considering the seminal role of customers’ communicative behavior (such as CADB) and observing that the available literature on CSR did not provide a sufficient explanation of how CSR-related communication on social media for a brand can influence CADB, the preliminary purpose of carrying out this investigation is to explore this relationship in a social media context. Second, and most importantly, we want to explore the above-proposed relationship in a service industry context where the communicative behavior of customers is more important than physical products because, unlike physical products, prior testing or experience is not possible in services [28,29,30]. Hence, the human dependence on the services segment makes the role of CADB more important. Especially, we want to test this relationship in the banking segment of a developing country (Pakistan) because the banking segment faces the issue of competitive convergence [31] due to its standardized nature of service delivery [32]. This standardization makes it very challenging for a bank to differentiate from the rest of the crowd. In this respect, we argue that the positive communicative behavior of customers such as CADB, as an outcome of CSR-related activities on social media, can well position a bank into competition. Therefore, it will be worthwhile to investigate the CSR–CADB relationship in the context of banking services.

Possibly, social scientists such as Glavas [33] and Aguinis and Glavas [34] were the first to spot the importance of different psychological aspects of human psychology that explain why individuals are expected to be engaged in a specific behavior in a CSR context. In specific, the inclusion of different psychological factors as mediators in a CSR framework provides a better justification for certain behavioral intentions of individuals. Especially, Glavas proposed that the debate already existed that CSR could influence the behavioral intentions of individuals, but how and why such relationships exist is only possible when we explain a specific relationship in the light of mediator(s). Perhaps this is one of the reasons why there is a recent surge in the literature emphasizing the importance of different mediators to explain the underlying mechanism of a certain behavioral aspect of human psychology. Extending this debate, we propose that customer–company identification (CCI) as a psychological factor of customer psychology can explain the underlying mechanism of why CSR-related communication on social media derives CADB. Of course, we are not the first ones to indicate the mediating role of CCI in influencing customer behavior in a CSR context because plenty of researchers have already mentioned the mediating role of CCI [35,36]. However, we believe we are the first ones to spark this discussion from the perspective of a CSR–CADB relationship in the prevalence of social media.

Similarly, an increasing body of literature has recently acknowledged the essential role of human emotions in influencing/shaping different individual outcomes. For example, Tsoy et al. [37] showed how communities perceive COVID-19 threats and risks, and how social media has the potential to reinforce ‘stay at home’ intentions by stimulating emotional regulation through intrinsic and extrinsic incentives. Similarly, researchers such as He et al. [38] also mentioned that emotions are of great importance to shape different attitudinal and behavioral outcomes of individuals.

In particular, the extant researchers believed that individual emotions have a seminal role in determining different behavioral aspects of customers, including, but not limited to, loyalty and CADB [39]. Possibly, this is why the literature in the recent past has largely focused on studying human emotions and how different emotions can drive a certain behavioral aspect of customer psychology [40,41]. Realizing the essential role of emotions, companies in the current era want to develop emotional relations with their interest groups, including customers. Specifically, organizations know that shared values, for example, pro-social activities, can generate an emotional pull on the customers’ part. In this regard, individuals’ feelings of gratitude (an emotional perspective), as a moderator, recently became a part of academic discussion. Specifically, researchers highlighted gratitude’s moderating role in sparking different individual outcomes [42,43]. Research also shows CSR’s role in sparking emotional feelings among individuals [44,45,46]. However, the role of CSR-related communication on social media to influence gratitude and how gratitude moderates the mediated relationship between CSR and CADB via CCI has not been discussed previously. Therefore, we tend to investigate this moderating role of gratitude in a social media context.

In a nutshell, our research intends to enrich the available literature in the following ways. First, our study tends to spark the discussion on CADB from a perspective of CSR, which has remained an under-investigated area. As mentioned above, most studies on CSR-related research streams were carried out with different behavioral perspectives other than CADB. For instance, loyalty [47,48], satisfaction [49], and purchase preference [50]. Considering the important role of the communicative behavior of customers in influencing/shaping the behavior of other customers, it is important to investigate the above-proposed relationship. Second, this study proposes a robust theoretical model to explain the CSR–CADB relationship. The reason for this robustness lies in the logic that our study is one of the few that simultaneously considers the mediating and moderating roles of CCI and gratitude to understand how and why CSR-related communication on social media can convert customers into advocates for a specific banking services organization. Third, our study intends to contribute to the existing body of literature related to CSR in the context of social media, especially from the standpoint of CADB. In this respect, limited studies exist on the relationship between CSR and CADB [51,52]. However, in these mentioned studies, the perspective of social media as an effective communication forum was not discussed. Finally, this study initiates the discussion on CSR in the context of a developing country, which is still sparse compared to developed countries. The literature argues that CSR is a context and culture-specific construct and, therefore, a universal customer orientation is almost impossible. For more detail, we refer to the study of Glaveli [53]. Similarly, developed and developing nations are dissimilar in many ways, and therefore, studies on CSR from developed countries are less likely to reflect developing countries’ perspectives. Therefore, separate studies are required in the context of developing countries.

## 2. Literature and Theoretical Underpinning

We base the theoretical discussion of this study on the theory of social identity (SI), perhaps one of the most employed theories by behavioral scientists to explain the logic of a particular individual behavior in different contexts. The polish psychologist Henri Tajfel was the person who initially proposed this theory during the 1970s [54]. Realizing the potential of this theory to explain the underlying logic of different human behaviors in different social contexts, many social scientists in the domain of human psychology have used this theory to date [55,56]. Originally, Henri Tajfel conceptualized this theory by proposing that an individual’s self-concept is partially derived from a particular social group to which an individual belongs. This theory further contends that once identified with a social group, an organization, for example, individuals feel their social group to be worthier than any other group and are expected to put forth every effort which can boost the social image of their group (for example, the ethical image in the current context). The fundamental tenets of this theory conceptualize that a particular person is expected to develop a strong social identity with a social group for which they think such identification with the group may induce their self-esteem. Projecting the crux SI in this study, we propose that the socially responsible image of a particular banking services organization is something that attracts contemporary customers to be identified with a bank of such kind that not only considers its own professional growth but also attempts to benefit all stakeholders as a part of its CSR strategy. Considering the rising expectation of contemporary customers for businesses to be engaged in different pro-social activities for the larger benefit of the ecosystem and society, an organization’s CSR-related communication on social media can certainly provide customers with a justified reason why they should identify themselves with an ethical organization. Perhaps this is why a plethora of previous CSR scholars have used SI as an underpinning theory in different individual surveys [57,58,59]. Even the marketing communication literature has recognized the potential of SI to explain a particular behavior of customers from a CSR perspective [60,61].

In line with the basic conceptualization of SI, we propose that CSR is an enabler to influence various customer outcomes, including loyalty [62], satisfaction [63], purchase intentions [64], and others. Even the literature informs us that there is a seminal role of CSR policies of an organization to influence various forms of customers’ extra-role behaviors positively. For example, it is well established in the literature that CSR engagement of brands to benefit society and the biosphere can positively influence customers’ citizenship behavior [65,66]. Similarly, scholars have emphasized that an ethical organization’s well-planned and well-executed CSR strategies can influence customers’ pro-social behavior [67,68]. From the perspective of communicative behavior, the recent literature acknowledges the essential role of CSR in fostering different forms of customers’ communicative behaviors, including word of mouth [69,70], referrals [71], and recommendations [72]. Even discussion exists on how CSR-related communication on different digital media platforms can escalate the communicative behavior of customers [73]. In line with this literature stream, we argue that CADB is also a form of communicative behavior by the customers for a particular brand. In this regard, when customers note the socially responsible commitment of a brand on social media, they strongly identify themselves with such ethical brands and not only prefer them in their purchase decisions but also promote such brands among their social circles by acting as brand advocates. Hence, we propose:

**H1.** 
*The manifestation of CSR-related communication of an organization on social media positively influences CADB.*


Similarly, another important psychological factor related to customer emotions is customer–company identification (CCI), which is defined as a customer’s psychological feeling of belongingness with an organization [74]. The literature defines CCI as an emotional aspect of individual psychology in which a particular individual shows belongingness for a focal agent, which is an organization in the current study [74]. Extending this definition of CCI with the theme of this study, we argue that the socially responsible engagement of an ethical organization creates an emotional bonding in the customer–company relationship. To be precise, in response to CSR, customers show a higher level of emotional feelings to be the buyers of an ethical organization [75]. Additionally, the above point has largely been emphasized in the recent literature that contemporary customers are more concerned about sustainable ecosystems and the biosphere when choosing a brand. In specific, customers of this digital age are not only concerned with the consumption of a product or a service, but they are also equally concerned about how their consumption decision impacts the ecosystem of this planet [76]. This clearly highlights why modern customers prefer to be buyers of sustainable brands and consider responsible brands to develop an emotional attachment [48].

Explaining this phenomenon further, Scott and Lane [77] indicated that the emotional bonding between customer and company creates an emotional pull among the customers, ultimately promoting them to a higher level of CCI. In particular, it is mentioned in the literature that the CSR engagement of a brand creates an emotional bond with different interest groups, including customers, which then increases their level of CCI [77].

To this end, the manifestation of social media further helps a brand to develop an emotional relationship by communicating with the customers about different CSR policies to the larger interest of society and all stakeholders [2]. From the standpoint of CCI, an increasing body of literature has indicated that companies can use social media to effectively communicate with their customers regarding CSR to foster their CCI [78,79]. The interactive connectivity of a brand with customers on a certain social networking website is something that actively engages customers with a particular brand [80]. Moreover, the past literature indicates that CSR positively influences CCI, which mediates between CSR and customer behavior [35,36]. Additionally, we are in line with the literature stream in which CCI has been identified as an important enabler to shape/influence various customer outcomes [81,82]. Hence, we propose that CSR communication on social media not only influences CCI, but there is a mediating role of CCI to influence CADB. Therefore:

**H2.** 
*The manifestation of CSR-related communication of an organization on social media positively influences CCI.*


**H3.** 
*There is a mediating role of CCI between CSR-related communication on social media and CADB.*


Whereas prior research on relationship marketing has strongly emphasized building an economic relationship with customers in which the focus was to establish a relationship with customers based on economic benefit (for example, discounts) [83,84], a recent body of literature argues that such relationships are short-lived and suggest interpersonal relations are long term in nature [85,86]. The reason why modern companies strongly emphasize building an interpersonal relationship with their interest groups, such as customers, lies in the fact customers feel a particular company wants to establish a relationship with its customers beyond the economic aspect [87]. Unlike the traditional relationship strategies, which offered a weak and short-lived value proposition, the recent relationship marketing literature emphasized heavily building long-lasting customer–company relationships. This highlights the importance of an emotional perspective to engage customers with a particular brand that provides longevity in a customer–company relationship [88,89].

In this respect, the role of gratitude as a psychological factor in influencing the positive emotional feelings of customers has recently been discussed in the literature at different levels [90,91]. By definition, gratitude is an other-directed volunteer emotional aspect of human psychology as a result of the recognition of some particular benefit received from others [92]. The literature highly rates gratitude in building emotional customer–company relationships [93,94]. Indeed, the manifestation of gratitude in a relationship further solidifies it by strengthening a particular buyer–seller relationship on the horizon of emotions. Scholars believe that gratitude is important in solidifying a relationship between customers and a company [95]. Specifically, the role of gratitude as a moderator has been discussed previously [42,96]. From the perspective of relationship marketing and communication, gratitude’s role in influencing a customer’s emotional aspects has been debated. Even the conditional indirect role of gratitude has been emphasized in the recent literature [42,97].

To this end, the past literature informs us that the CSR activities of a company enhance customer gratitude significantly [98,99]. Hence, we argue that when customers see CSR-related communication of a particular organization on social media, they perceive it as a benefit received from others (a socially responsible company) for all stakeholders, including customers. In response, customers develop a positive emotional state to support an ethical organization, which positively influences the mediate relationship between CSR and CADB through CCI. Therefore:

**H4.** 
*Gratitude moderates the mediated relation of CSR and CADB via CCI.*


The conceptual model of this study is given in Figure 1. 

## 3. Methodology

### 3.1. Study Sector and Data Collection

Pakistan’s banking sector is the target segment for this investigation. Mainly the State Bank of Pakistan (SBP) is the regulating body that devises different regulating mechanisms and standard operating procedures (SOPs) for all banking services organizations. There are two major banking streams in this South Asian nation comprising regular conventional banking (interest-based) and Islamic banking (interest-free). Nonetheless, the conventional banking stream is the leading one in Pakistan, holding almost 80% of the total banking industry’s share [100]. In conventional banking, Habib Bank Limited (HBL) is the leading bank, whereas Meezan bank (MB) is the largest representative of Islamic banking. Because the founding objective carrying out this research was to investigate how CSR communication of a banking institution on social media influences the advocacy behavior of interest groups, especially customers, we first identified the banking institutions with different CSR activities. Moreover, we also identified the banks actively using different social networking websites to communicate their CSR activities with the external community. In this regard, we discovered that the larger banks in both banking streams not only had specific CSR programs but also used social media to interactively communicate with the customers. Therefore, we selected the five largest banking organizations operating in Pakistan: HBL, National Bank of Pakistan (NBP), MB, Bank Alfalah (BA), and MCB bank.

For data collection, we approached various banking customers in Lahore and Karachi while they were leaving a particular banking branch or were found around a bank’s ATM area. This data collection strategy was considered due to some reasons. First, this strategy is useful because it does not interrupt the banking operations of any bank. Second, this strategy enabled us to collect data from real banking customers. Lastly, this strategy helped us in approaching customers randomly of all ages. Raza et al. [81] and Sun et al. [101] have also used the same in their recent studies. We used a printed questionnaire to collect the responses from various banking customers. Specifically, customers were screened out by asking about their social media engagement and basic CSR knowledge. The variables’ statements were taken from different published and authenticate resources. Considering the ethical concern in this study, we employed major Helsinki Declaration guidelines [102,103]. To be precise, the data collection activity took two months (February to April 2022). Knowing the fact that survey research does not produce a 100% response rate, we disseminated 500 questionnaires, and we were successful in receiving 323 filled questionnaires. After data scrutiny, 21 responses were deleted because these responses either were partially completed or were outliers. Specifically, 14 questionnaires contained missing information and were identified as invalid. Among these 14 questionnaires, the highest filling percentage of a questionnaire was 55%, meaning that 45% of responses were left unfilled by the respondent. Further detail has been given in Table 1 and Table 2.

### 3.2. Measures

All responses were gathered on a 5-point Likert scale. Experts from the field and academia were requested to judge the suitability of all items in relation to this study [104,105,106,107]. Indeed, we took five items to quantify the CSR-related communication of a brand on social media from the study of Fatma et al. [108] which was originally based on the studies by Brown and Dacin [109] and Klein and Dawar [110]. Specifically, these items were modified in the context of social media. For example, one sample item was, “When I see CSR-related communication of my bank on a social networking website, I feel that this bank is socially responsible”. To quantify the CCI, we adapted five items from the work by Eberle et al. [111], which included a sample item, “I have a sense of connection with my bank.” The variable of gratitude-GA was quantified by using a three-item scale from the study of Kim and Park [99]. One item from this scale was, “I feel grateful for my bank’s effort to contribute to our society”. Lastly, the variable of CADB was measured by using four items developed by Melancon et al. [112]. A sample item from this scale was, “I would defend my bank on social media to others if I heard someone speaking poorly about it”. To ensure reliability, we checked the Cronbach alpha value (α) of each variable. Generally, a frequently referred value of α is 0.70, which is based on the recommendations of Nunnally [113]. Several other authors have also suggested the same [114,115]. In this respect, we observe α = 0.83, 0.82, 0.72, and 0.80 for CSR, CCI, GA, and CADB, respectively Appendix A Table A1.

### 3.3. Social Desirability and Common Method Bias

Dealing with social desirability and common method variance (CMV) are significant, especially when the data were gathered in a single wave from a particular respondent (such as in this study). Therefore, we paid significant attention to these issues theoretically and statistically in devising different methodology-related protocols. Theoretically, we cleared all customers who participated in this survey that there is no bad or good choice for a particular response. Moreover, the respondents were cleared about the importance of their true response in drawing a genuine conclusion for this study based on their provided information. Similarly, we randomly presented the items of a variable to the respondents so that they were less likely to develop any sequence in answering a particular question.

Statistically, we performed a common latent factor test (CLT) to see if this study’s dataset faces any CMV issues. For this purpose, we drew two particular measurement models using AMOS software. One measurement model was the original four-factor model without any manifestation of a common latent factor (CLF). In the other model, we included a CLF in the measurement model. The statistical output of the CLT revealed that the standardized regression weights of both models did not produce any significant variation (>0.20), implying that the inclusion of the CLF in the measurement model did not create any significant variance [116,117,118]. Hence, CMV was not critical in this study. Socio-demographic information is given in Table 3.

## 4. Results

### 4.1. Preliminary Statistical Analysis

In the preliminary statistical analysis of the data, we ran different statistical tests mainly to establish the validity and reliability of the variables included in this analysis. In this regard, we tested an important validity type, convergent validity, by estimating the average variance extracted (AVE). Normally, the established criterion for significant validity is assumed in a case where the AVE is greater than 0.5 [119,120,121,122]. In the current analysis, we revealed no variable’s AVE was below this threshold level. To be precise, the AVEs varied from 0.55 (GA) to 0.61 (CADB). This statically confirmed that convergent validity was significant for CSR, CCI, GA, and CADB.

Similarly, we revealed that composite reliability was significant because no variable’s value was less than 0.7, which is a standard threshold level for accepting the composite reliability of a variable [123,124]. Particularly, composite reliability values were between 0.78 (GA) and 0.87 (CSR). These results provide significant statistical ground to accept composite reliability for all variables. Table 4 includes further information. Moreover, the measurement model is given in Figure 2.

In the next level, we verified the goodness of the model fit of our theoretical model by drawing four measurement models and comparing them with the help of different model fit indices and other model fit indicators. Specifically, we developed these measurement models with different compositions; however, the original theoretical model (model 1) was built with the original composition (CSR = 5 indicators, CCI = 5 indicators, GA = 3 indicators, and CADB = 4 indicators). We have provided further detail in Table 5. The goodness of model fit comparison indicated that measurement models with different compositions produced mixed results. For example, a one-factor model poorly explained data in all respects (RMSEA = 0.129, *χ*^2^/*df* = 8.69, GFI = 0.59, TLI = 0.55, IFI = 0.57, CFI = 0.60). Model 2 and model 3 produced mixed results in which some values were reasonably good (for example, TLI = 0.83 and *χ*^2^/*df* = 4.59). Nonetheless, only model 1 was identified as a measurement model with excellent results indicating that there was a good fit between theory and statistical data (RMSEA = 0.068, *χ*^2^/*df* = 2.58, GFI = 0.92, TLI = 0.91, IFI = 0.94, CFI = 0.94).

We also observed inter-variable associations in different cases by observing the correlational (*r*) values. This assessment was important to establish whether the hypotheses’ theoretical statements were in agreement with the statistical results. We realized that the *r*-values were significant and positive in all cases. Specifically, these values were between 0.37 (CCI <=> CADB) and 0.61 (CSR <=> GA). These values were significant and provided initial statistical support to the hypotheses’ statements, especially for H1, H2, and H4. We also examined divergent validity, which we reported here in Table 5. While convergent validity is essential to see whether the items of a particular variable converge on it or not, divergent validity provides statistical evidence that the items of a variable are not identical compared with the items of other variable(s). In this regard, we found that the divergent validity for all variables was significant (please see the bold values in Table 6).

### 4.2. Main Statistical Analysis

In the main statistical analysis, we evaluated the hypothesized associations of variables by using structural equation modeling (SEM). IBM-SPSS and AMOS software were used to develop a structural model. Indeed, SEM is an advanced data analysis option that is valuable, especially for analyzing complex models [120,121,125,126]. Before developing the structural model, some important estimation measures were taken. For example, the normality of data was assured by observing the skewness and kurtosis values of all variables. Similarly, CSR and GA were mean-centered. Additionally, an interaction term (CSR_X_GA) was also created to see the conditional indirect effect of GA. The PROCESS-Macro [127] model 7 was followed to generate different mathematical expressions in AMOS by pursuing a user-generated estimated option. Lastly, a larger bootstrapping sample of 5000 was used to see the significance of mediation and moderation effects [128]. For the convenience of readers, we have summarized the structural analysis results in Table 7.

For more simplicity, we first presented the direct effect results of our structural model (H1 and H2). In this vein, we revealed that H1: CSR→CADB was significant because the beta value = 0.49 was positive and significant (*p* < 0.05 and no CI value included a zero point). This statistically confirmed that the theoretical statement of H1 was statistically true. A similar conclusion may be drawn for H2. Second, the mediating effect of CCI betwixt CSR and CADB was also significant (CSR→CCI→CADB = 0.31, *p* < 0.05), thereby indicating that H3 was significant. Last, we investigated the condition effect produced by GA betwixt CSR and CCI at different levels of GA (for example, at mean and ±1 standard deviation). The statistical evidence in this stage of structural analysis confirmed that GA moderates significantly between CSR and CCI at all three levels. Moreover, the conditional indirect effect of GA was also significant (beta value = 0.38, *p* < 0.05). This statistically validated that H5 was significant. Figure 3 shows the structural model diagram.

## 5. Discussion

This study was conducted mainly to see whether CSR-related communication on social media by a particular bank can determine an important communicative behavior of customers, CADB. In this context, the statistical analysis showed that CSR significantly predicts CADB (beta = 0.49), confirming that the theoretical statement of H1 was statistically valid. This statistical finding can be related to the theoretical discussion in line with this study’s objective. Specifically, our results show that the manifestation of CSR communication by a bank on social media positively impacts their behavior, especially CADB. This is in line with the existing theoretical debate on CSR and individual behavior. When customers observe that a particular banking services organization devises different CSR plans to benefit all stakeholders, including the ecosystem and biosphere, they feel that this bank is socially responsible and takes care of every stakeholder. Being included in the list of important stakeholders, customers are also beneficiaries of CSR, hence, they positively respond back to the ethical organization by showing their extra support and engagement. Contemporary customers, as brand advocates, not only stand with ethical brands but are also expected to defend their beloved brands, due to their social engagement, against detractors. Particularly, in line with the basic tenets of SI, a particular customer tends to develop a strong social identity with a social group (a bank) for which they feel such identification will be helpful in enhancing their self-esteem. In this regard, the socially responsible image of a particular bank is something that attracts contemporary customers to be identified with a bank of such kind, which not only considers its own professional growth but also attempts to benefit all stakeholders as a part of its CSR strategy. Hence, a bank’s CSR-related communication on social media can certainly provide customers with a justified reason why they should identify themselves with an ethical bank. Especially from an impression management perspective, companies can improve their sustainability image by effectively communicating their CSR activities with external stakeholders on social media. Such impression management perspective is helpful for companies to grab stakeholders’ support [129]. In a nutshell, customers perceive CSR activities by a bank as other-provided benefits and respond back positively by developing strong associations and by displaying positive communicative behaviors, including CADB. Previous research also confirms that CSR positively influences CADB [89,130,131].

Similarly, our research confirms that individual emotions play a role in explaining the underlying mechanism of why individuals are engaged in a particular behavior. In this regard, our research confirms that CCI, as an emotional aspect of human psychology, significantly explains why the CSR activities of a banking organization influence CADB (beta = 0.31). This confirms that H2 was statistically significant and hence accepted. In specific, the socially responsible image of a banking organization inculcates positive emotions among customers, which ultimately becomes the reason for a strong CCI on the part of customers.

Customers in this current age of digital technology show a greater concern for sustainable ecosystems and the biosphere before considering a brand in their purchase decisions. To be precise, customers are not only concerned with the consumption pattern of a product or a service, but they are also equally concerned about how their consumption decision helps this planet from the perspective of sustainability. In response to CSR, the emotional relation between customer and company creates positive feelings among the customers, ultimately promoting them to a higher level of CCI. The manifestation of social media further helps an organization, for example, a bank, to develop an emotional relation by communicating with the customers about different CSR policies to the larger interest of society and all stakeholders. Banking organizations can use social media to effectively communicate with their customers regarding CSR to foster their CCI. The interactive connectivity of a bank with customers on social media actively engages customers, creating an opportunity for companies to foster CCI. Hence, we confirm the mediating role of CCI betwixt CSR and CADB and, therefore, our H3 was also verified in light of statistical evidence. This is in line with the previous literature [35,36], which posited that CSR could influence various individual outcomes.

Lastly, our study also confirms that the manifestation of gratitude further buffers the mediated relationship of CSR and CADB via CCI (beta = 0.38). This provides statistical evidence for accepting the theoretical statement of H4. Indeed, the manifestation of gratitude in a relationship further solidifies it by strengthening a particular buyer–seller relationship from an emotional perspective. From a relationship marketing perspective, gratitude’s role in influencing a customer’s emotional aspects is critical because gratitude, stemming from CSR, further elevates customers’ positive feelings in the form of CCI, which then improves CADB. Hence our study confirms the conditional indirect role of gratitude which is in line with the previous literature [42,97].

### 5.1. Implications

#### 5.1.1. Implications for Theory

Our research significantly sparks the existing debate on the CSR–customer relationship by offering different theoretical perspectives. In the first vein, our study is one of the limited investigations on CADB from the perspective of CSR, which has remained, to date, an under-investigated area. In this regard, most CSR scholars produced studies indicating that CSR can influence customer loyalty [47,48], satisfaction [49], and brand purchase preferences [50]. Considering the important role of the communicative behavior of customers to influence/shape the behavior of other customers, it was important to investigate the above-proposed relationship. In the second vein, our study proposes a more robust theoretical model to explain the CSR–CADB relationship because our study is one of the fewest investigations that attempt to provide a better justification of how and why CSR relates to CADB by simultaneously considering the mediating and moderating roles of CCI and gratitude to understand the underlying mechanism of the CSR–CADB relationship. In the third vein, our study intends to contribute to the growing body of literature related to CSR in the context of social media, especially from the standpoint of CADB. In this respect, limited studies exist on the relationship between CSR and CADB [51,52]. However, in these mentioned studies, the perspective of social media as an effective communication forum was not discussed. Last but not least, this study initiates the discussion on CSR in the context of a developing country which, compared to developed countries, is still sparse. We have already mentioned that CSR is a context and culture-specific construct; therefore, a universal customer orientation is not possible, as indicated by Glaveli [53]. Similarly, developed and developing nations are dissimilar in many ways and, therefore, studies on CSR from developed countries are less likely to reflect the perspective of developing countries. Therefore, separate studies are required in the context of developing countries.

#### 5.1.2. Implications for Practice

Considering the rising traction of social media in the recent past and considering the important role of customers in the success of a business, our study offers significant practical contributions, especially to the banking segment of Pakistan. To this end, in light of statistical evidence, it is established that communicating CSR-related information on social media can generate various positive outcomes for a banking organization. Especially from the perspective of communicative behavior, CSR-related communication has a seminal role in sparking CADB. Customers feel elevated to be the buyers of a socially responsible organization because they believe that in being the purchasers of an ethical banking organization they are also contributing to the larger benefit of society and the biosphere. Specifically, CSR-related communication on social media not only enhances customer loyalty preferences but also promotes their beloved brands on social media by showing advocacy behavior. The above insight has a very special relevance to the banking industry because, considering the standardized procedure to serve customers, this industry faces the issue of competitive convergence in which every banking organization has the same or less differentiated set of skills to grab the loyalty of customers. To this end, this study shows that if a certain bank crafts effective CSR strategies and actively communicates such CSR actions with the customers on social media, it can help a bank find a stable point of competitive advantage in the form of customers as brand advocates.

Another practical insight of our study is to highlight the seminal role of customer emotions in influencing their behavior, especially CADB. In this respect, the literature already acknowledges that emotionally charged customers bring various advantages to a certain brand. Customers with a higher level of emotional engagement with a certain brand are less likely to quit a brand; even in difficult times they stand by their preferred brand due to their emotional relationship with such brands. To this end, our results indicated that there is a role of customer emotions in the form of CCI, which motivates them to strongly identify themselves with a bank due to its socially responsible behavior. Ultimately, the emotional pull generated by a brand due to its social responsibility engagement converts customers into brand advocates. Because social media provide an interactive communication forum, customers not only interact with a particular brand on social media but also tend to share their relationship experience with a brand in their social circles. Hence, a certain banking organization can promote CADB among customers using social media as an effective CSR communication strategy.

### 5.2. Limitations and Future Research Suggestions

Considering this study’s different significant theoretical and practical implications, one should not mark this investigation error-free. Indeed, there are some potential limitations in this study which we want to highlight with possible suggestions for future researchers. First of all, this study collected data from two metropolitan cities in Pakistan. Due to different resources and time constraints, we did not include other cities in this survey, and hence the geographical concentration of this study is something that may serve as a potential limitation. For future studies, we recommend including more cities to deal with this limitation. Similarly, though the mentioned relations were all significant, we still feel that the theoretical framework of this study may be enriched by incorporating more variables as mediators and moderators. For example, it may be interesting to see how the role of admiration and gratitude as moderators or mediators in future investigations may further spark CADB. Hence, it is suggested to include more variables in the theoretical framework of this study. Lastly, considering the context-specific and cultural-specific nature of CSR, it is hard to generalize the results of this study to other segments of an economy and other cultures, though we believe in similar cultures such as Bangladesh or India our study may reflect the same findings. However, we still recommend that future studies produce a comparison of different service segments and cultures. For example, the comparison of the hospitality sector with banking may be important in future studies because the hospitality sector also faces the challenge of competitive convergence like banking.

## 6. Conclusions

The banking industry is at a crossroads. Rising rivalry creates a challenge for banking administrators on one side. On another side, the standardized operating procedure raises the barrier to hold, build, and sustain loyal customers in this segment. This indicates that finding a stable competitive position in this sector is not easy for a bank. Moreover, dynamic business environments, cutting-edge technologies, and many other factors force banking administrators to build meaningful relationships with their customers. Undoubtedly, without the support of customers, it is almost impossible for a bank to be successful. In this respect, we suggest banking administrators carefully develop different CSR strategies and communicate them with the customers using different social media forums because, without meaningful communication of CSR activities with customers, it will be difficult for a bank to grab the opportunity to convert customers into brand advocates. Similarly, the administration of a specific bank is suggested to align different marketing communication strategies by incorporating the emotional perspective of human behavior. Such emotional alignment should also reflect in CSR-related communication on social media because emotionally elevated customers, in response to CSR, are expected to build a strong customer–company relationship and share their positive experience with a brand with their social circles. Indeed, banking administration needs to realize that shared values (CSR, for example) may be critical for effectively generating emotional connections that drive customers’ loyalty and advocacy. All in all, if rising rivalry and limited differentiation options are the biggest challenges in the banking segment, effective CSR communication on social media is a way forward for banking organizations.

## Figures and Tables

**Figure 1 behavsci-13-00126-f001:**
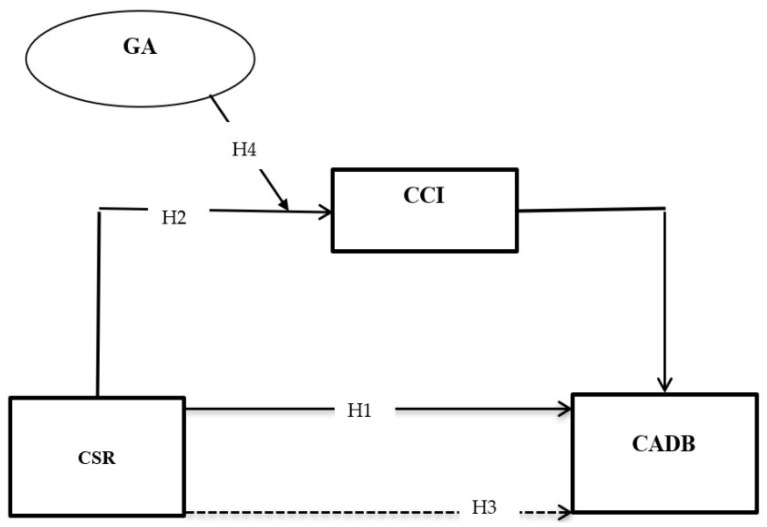
Theoretical framework.

**Figure 2 behavsci-13-00126-f002:**
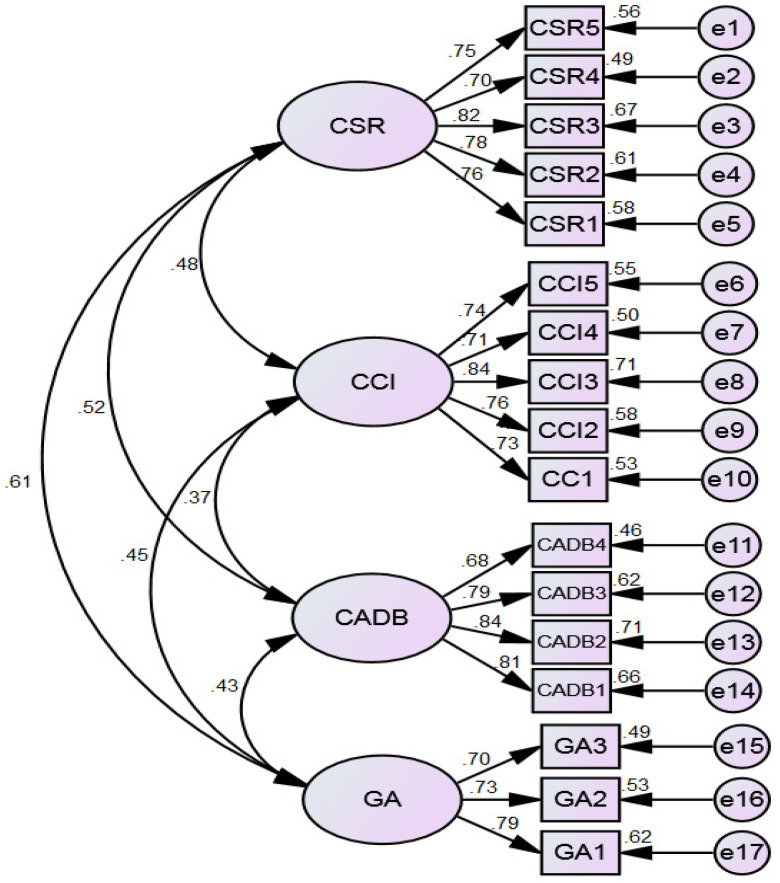
Measurement model.

**Figure 3 behavsci-13-00126-f003:**
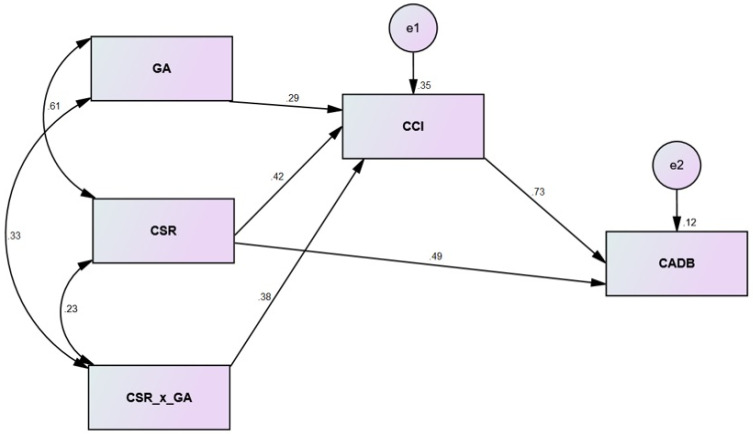
Structural model.

**Table 1 behavsci-13-00126-t001:** Data cleaning, outliers, and response rate.

Distributed	Returned	Unreturned	Removed	Outliers	Final
500	323	177	21	07	302
-	64.6%	35.4%	06.5%	33.3%	60.4%

**Table 2 behavsci-13-00126-t002:** Outliers.

Case No.	Mahalanobis d-Squared	*p*1	*p*2
20	19.097	0.000	0.021
75	19.097	0.000	0.000
130	19.097	0.000	0.000
186	19.097	0.000	0.000
240	19.097	0.000	0.000
7	14.579	0.001	0.000
62	14.579	0.001	0.000
20	19.097	0.000	0.021

**Table 3 behavsci-13-00126-t003:** Socio-demographic information.

Demographic	Frequency	%
Gender		
Male	199	65.89
Female	103	34.11
Age		
18–25	46	15.23
26–30	87	28.81
31–40	69	22.85
41–45	57	18.87
Above 45	43	14.24
Education		
Intermediate	33	10.93
Graduate	54	17.88
Master	184	60.93
Higher	31	10.27

**Table 4 behavsci-13-00126-t004:** Summary of initial analyses.

	λ	λ^2^	E-Variance
CSR	0.75	0.56	0.44
AVE = 0.58	0.70	0.49	0.51
CR = 0.87	0.82	0.67	0.33
∑ λ^2^ = 2.91	0.78	0.61	0.39
Total items = 5	0.76	0.58	0.42
CCI	0.74	0.55	0.45
AVE = 0.57	0.71	0.50	0.50
CR = 0.87	0.84	0.71	0.29
∑ λ^2^ = 2.87	0.76	0.58	0.42
Total items = 5	0.73	0.53	0.47
GA	0.70	0.49	0.51
AVE = 0.55	0.73	0.53	0.47
CR = 0.78	0.79	0.62	0.38
∑ λ^2^ = 1.65	-	-	-
Total items = 3	-	-	-
CADB	0.68	0.46	0.54
AVE = 0.61	0.79	0.62	0.38
CR = 0.86	0.84	0.71	0.29
∑ λ^2^ = 2.45	0.81	0.66	0.34
Total items = 4	-	-	-

Notes: λ = Item loadings, CR = composite reliability, ∑ λ^2^ = sum of the square of item loadings, E-Variance = error variance.

**Table 5 behavsci-13-00126-t005:** Model fitness.

Model	*χ*^2^/*df* (<3)	Δ*χ*^2^/*df* *-*	RMSEA (<0.08)	GFI (>0.9)	TLI (>0.9)	IFI (>0.9)	CFI (>0.9)
1	2.58	_	0.068	0.92	0.91	0.94	0.94
2	4.59	2.43	0.073	0.85	0.83	0.85	0.86
3	5.82	1.44	0.080	0.79	0.77	0.79	0.81
4	8.69	1.59	0.129	0.59	0.55	0.57	0.60

Note: 1 = four-factor model, 2 = three-factor model by combining CSR + CCI into one factor, 3 = two-factor model by combining CSR + GA + CCI and CADB, 4 = one-factor model by combining CSR + GA + CCI + CADB.

**Table 6 behavsci-13-00126-t006:** Correlations and discriminant validity.

Variable	1	2	3	4
1	**0.76**	0.48	0.61	0.52
2	(3.29 _m_, 0.65)	0.76	0.45	0.37
3		(2.96, 0.53)	0.74	0.43
4			(3.49, 0.74)	0.78
				(2.99, 0.60)

Notes: values in parenthesis = mean and standard deviation, bold values = discriminant validity, *p* < 0.001, 0.05. 1 = CSR, 2 = CCI, 3 = GA, 4 = CADB.

**Table 7 behavsci-13-00126-t007:** Structural analysis.

Hypotheses	Estimates (SE)	*t*-Value	*p*-Value	CI
(CSR→CADB) (CSR→CCI)	0.49(0.057) 0.42(0.054)	8.59 7.78	**** ****	0.19, 0.56 0.24, 0.59
Indirect effect (CSR→CCI→CADB)	0.31(0.039)	7.95	****	0.11, 0.47
Conditional effect of GA betwixt CSR and CCI				
−1SD	0.036(0.014)	-	-	0.12, 0.19
At mean	0.040(0.018)	-	-	0.10, 0.16
+1SD	0.046(0.021)	-	-	0.23, 0.29
The conditional indirect effect of IM between CSR→CCI→CADB	0.38(0.029)	13.10	****	0.27, 0.44

Notes: CI = 95% confidence interval with lower and upper limits. Model fit indices: *χ*^2^/*df* = 2.52, RMSEA = 0.067, GFI = 0.93 TLI = 0.92, IFI = 0.94, CFI = 0.95, **** = significant *p*-values.

## Data Availability

Data will be made available by contacting the corresponding author on a reasonable request.

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
