# Peer review of "The Role of CSR Information on Social Media to Promote the Communicative Behavior of Customers: An Emotional Framework Enriching Behavioral Sciences Literature"

_behavsci, 2023, doi:10.3390/bs13020126_

Round 1

Reviewer 1 Report

1. This study investigates the association between CSR-related communication of a banking organization on social and CADB. Also highlights the role of human emotions in behavior formation by considering the mediating and moderating role of customer-company identification (CCI) and gratitude.The results of this study should be helpful in banking practice but there are still some writing problems for reference.

2. Please express the hypothesis in Figure 1 to facilitate cross-reference between the text and the figure 1.

3.In the section 3.1 on the collection of data, please add an explanation of the principle of deleting invalid questionnaires during the process of returning questionnaires. Although there are explanations in the text that are for partial incompleted or outliers, what are the principles for judging partial incompleted?

4.In section 3.2, it is recommended to list the reference sources, reliability and validity of each facet and item.

5.Although Table 6 shows the analysis results of the structural model, in order to facilitate the understanding of the analysis of the overall model, it is recommended to add the analyzed model diagram and mark the analysis results of Table 6 on the diagram.

6.In the discussion, in order to clearly understand whether each hypothesis has been verified, please clearly state in the content whether the various assumptions are true.

Author Response

1. This study investigates the association between CSR-related communication of a banking organization on social and CADB. Also highlights the role of human emotions in behavior formation by considering the mediating and moderating role of customer-company identification (CCI) and gratitude. The results of this study should be helpful in banking practice but there are still some writing problems for reference.

Response: Thank You!

2. Please express the hypothesis in Figure 1 to facilitate cross-reference between the text and the figure

Response: Thank you for the above comment. We have revised the figure 1 by following your suggestion. Hopefully, you will like revised Figure 1. Thanks again.

3.In the section 3.1 on the collection of data, please add an explanation of the principle of deleting invalid questionnaires during the process of returning questionnaires. Although there are explanations in the text that are for partial incompleted or outliers, what are the principles for judging partial incompleted?

Response: Well, the questionnaires were considered invalid or unanalyzable based on three particular principles. 1. Partially filled responses. In this regard if a questionnaire contains 90% responses by a respondent and 10% were left unfilled we still can include these responses in the final dataset. For this purpose we can give a neutral code to the unfilled responses (for example, 3 or 5 in five-point or seven point scales). 2. Outliers and 3. Responses with repeated ratings (for example, 1,1,1,1, or 2,2,2,2, in whole questionnaire).

However, in our case there were 21 responses which we treated as invalid. Regarding partially filled questionnaires, we observed that there were 14 questionnaires which were not fully filled (highest filled questionnaire showed around 55% rate, meaning that 45% responses were not filled). Regarding repeated responses, we luckily, did not observe any case. Lastly, 07 responses were treated as outliers.

We have also reflected the above discussion in the revised manuscript. Thanks again for the above feedback.

4.In section 3.2, it is recommended to list the reference sources, reliability and validity of each facet and item.

Response: All noted and thank you again for the above feedback. We have provided the reference sources along with the reliability values. Kindly see page 9, yellow highlighted text in the revised manuscript. Regards

5.Although Table 6 shows the analysis results of the structural model, in order to facilitate the understanding of the analysis of the overall model, it is recommended to add the analyzed model diagram and mark the analysis results of Table 6 on the diagram.

Response: The required diagram has been provided in the revised manuscript as Figure 2. Thanks

6.In the discussion, in order to clearly understand whether each hypothesis has been verified, please clearly state in the content whether the various assumptions are true.

Response: Thank You again for sharing with us the above valued thoughts. We have clearly stated in the main content of the revised manuscript that each hypothesis was verified. Kindly see the yellow highlighted text in the revised discussion part. Thanks

Reviewer 2 Report

1 The abstract needs improvements to be more concise. The intro to the abstract is long, and methods, sample are ommitied, as well as originality aspects.

2 Significance of CSR and use in a similar framework can be found it Influence of Virtual CSR Co-Creation on the Purchase Intention of Green Products under the Heterogeneity of Experience Value

3.  Please enrich your theoretical part with more literature that illustrates social media and networks importance in various spheres

For example like Continuous Influence-Based Community Partition for Social Networks. IEEE Transactions on Network Science and Engineering, 9(3), 1187-1197. doi: 10.1109/TNSE.2021.3137353

 Influence-Based Community Partition With Sandwich Method for Social Networks. IEEE Transactions on Computational Social Systems, 1-12. 

4. Provide more critical review from behavior and human nature perspective how it shapes behavior>Impact of Social Media, Extended Parallel Process Model (EPPM) on the Intention to Stay at Home during the COVID-19 Pandemic

 How Does Inequality Affect the Residents' Subjective Well-Being: Inequality of Opportunity and Inequality of Effort. Frontiers in psychology, 13, 843854. 

3.  Extensive check of grammar to be done

Example: lines 48-50 In this respect, 48 statistical evidence suggests that moreover that 4.5 billion people use different social networking websites

Lines 82-84 Albeit theoretical and empirical evidence exist to support social media’s seminal role to engaging customers with a brand, especially from a CSR perspective. However, there 83 exists a critical gap in this literature stream.

4.  Discussion part to include more references, especially when you are referring to the existing theories/literature. Also what are implications for companies / Substantial response or impression management? Compliance strategies for sustainable development responsibility in family firms. Technological Forecasting and Social Change(174). 

5.  Please include questionnaire and demographic statistics

6.  Were there any modification done in CFA and SEM in terms of correlations and removing items with low loadings?

7.  Please provide model fit indices for SEM

8. The figures are not appropriate. Figure 1 is a figure from software but without full variable names. Figure 1 should be of your own design.

Where are CFA and SEM figures? Figure 2, and 3?

9. The reporting of indicators in text should be done following APA standard.

Paper has potential but requires polishing and addressing all the points, indicating changes done in the paper in color or track changes.

Author Response

1 The abstract needs improvements to be more concise. The intro to the abstract is long, and methods, sample are ommitied, as well as originality aspects.

Response: Thank you for the above feedback and for indicating the above issue in our abstract. Following your worthy suggestions, we have revised our abstract. The revised version of our abstract is as follows.

Studies have shown that an organization's corporate social responsibility (CSR) activities affect customer behaviors, such as loyalty and satisfaction. In spite of this, the role of social media in informing customers about a brand's CSR activities and in fostering customer advocacy behavior (CADB) has been underexplored. To fill this knowledge gap, this study investigates the relationship between CSR-related communication of a banking organization and CADB. This study also examines how emotions such as customer-company identification (CCI) and gratitude as a mediator and a moderator. Using a self-administered questionnaire (n = 302), we collected data from banking customers. Hypotheses were evaluated by using structural equation modeling which revealed that CSR positively predicts CADB whereas there is mediating and moderating functions of CCI and GA. Theoretically, this study highlights the role of human emotions in behavior formation from the standpoint of social media. Practically, this study provides important insights to the banking sector’s administrators to realize the important role of CSR communication, using different social networking websites, for converting customers into brand advocates.

The same we have reflected in the main document. Thanks

2 Significance of CSR and use in a similar framework can be found it Influence of Virtual CSR Co-Creation on the Purchase Intention of Green Products under the Heterogeneity of Experience Value

Response: Thank you again for the above comment and for indicating the above important study. We have mentioned this important work in the introduction section during the revision. Best regards

  1. Please enrich your theoretical part with more literature that illustrates social media and networks importance in various spheres

For example like Continuous Influence-Based Community Partition for Social Networks. IEEE Transactions on Network Science and Engineering, 9(3), 1187-1197. doi: 10.1109/TNSE.2021.3137353

 Influence-Based Community Partition With Sandwich Method for Social Networks. IEEE Transactions on Computational Social Systems, 1-12. 

Response: Thanks for sharing with us the above valued perspective. We have enriched our introduction part by including more literature. In addition we have also reflected the crux of the above two studies in the revised version of our introduction. Hope you will like our revised efforts. Thanks

  1. Provide more critical review from behavior and human nature perspective how it shapes behavior>Impact of Social Media, Extended Parallel Process Model (EPPM) on the Intention to Stay at Home during the COVID-19 Pandemic

 How Does Inequality Affect the Residents' Subjective Well-Being: Inequality of Opportunity and Inequality of Effort. Frontiers in psychology, 13, 843854. 

Response: Following the suggestions of the worthy reviewer, we have enriched our theoretical discussion by adding the above studies. The added text has been highlighted in yellow for your kind perusal. Thanks again for the kind feedback on our work.

  1. Extensive check of grammar to be done

Example: lines 48-50 In this respect, 48 statistical evidence suggests that moreover that 4.5 billion people use different social networking websites

Lines 82-84 Albeit theoretical and empirical evidence exist to support social media’s seminal role to engaging customers with a brand, especially from a CSR perspective. However, there 83 exists a critical gap in this literature stream.

Response: All noted, and thank you for indicating the above grammar related issues. We have reconsidered our approach during the revision stage. Hopefully, our revised version will not include any of such grammar-related issue. Best wishes.

  1. Discussion part to include more references, especially when you are referring to the existing theories/literature. Also what are implications for companies / Substantial response or impression management? Compliance strategies for sustainable development responsibility in family firms. Technological Forecasting and Social Change(174). 

Response: Thanks again for the above comment. We have enriched our discussion section by including more references, especially, the crux of above study has been reflected in the revised discussion section. The added/modified text has been highlighted in yellow. Regards

  1. Please include questionnaire and demographic statistics

Response: The questionnaire has been provided in Appendix A and demographic detail have been included in revised Table 3. Thanks again.

  1. Were there any modification done in CFA and SEM in terms of correlations and removing items with low loadings?

Response: although we have been modifying our models previously as it is common in survey researches to observe some factors with lower loadings. However, fortunately this time our model showed good factor loadings in all respects. This is why we did not delete any item.

  1. Please provide model fit indices for SEM

Response: The required fit indices have been provided under Table 7 in the revised manuscript.

  1. The figures are not appropriate. Figure 1 is a figure from software but without full variable names. Figure 1 should be of your own design.

Where are CFA and SEM figures? Figure 2, and 3?

Response: We have redeveloped Figure 1 as our own. Moreover, CFA and SEM models have also been included in the revised manuscript as Figure 2 and Figure 3. Regards

  1. The reporting of indicators in text should be done following APA standard.

Paper has potential but requires polishing and addressing all the points, indicating changes done in the paper in color or track changes.

Response: We have modified the areas which were not according to APA. Hopefully, you will like it this time.

Thank you for the kind comments and positive evaluation. Best Wishes.

Round 2

Reviewer 2 Report

the paper was improved. But should be edited for redundancy issues such as 

"by pursuing the famous data analysis technique," structural equation modeling (SEM)

"As mentioned at the onset of this draft

and language ", an escalating number of companies"m etc.

Maybe CR and AVE equations are not required as those are not the main model representations of the study.

Author Response

The paper was improved. But should be edited for redundancy issues such as 

"by pursuing the famous data analysis technique," structural equation modeling (SEM)

"As mentioned at the onset of this draft

and language ", an escalating number of companies"m etc.

Response: Thank you again for the kind guidance and for positive evaluation. We have further improved the grammar related issues in our manuscript. Hopefully, this time such issues will not be evident in our work. Best Regards

Maybe CR and AVE equations are not required as those are not the main model representations of the study.

Response: All noted, we have removed both equations from the revised manuscript. Thanks again